# Could an Immersive Virtual Reality Training Improve Navigation Skills in Children with Cerebral Palsy? A Pilot Controlled Study

**DOI:** 10.3390/jcm11206146

**Published:** 2022-10-18

**Authors:** Roberta Nossa, Chiara Gagliardi, Daniele Panzeri, Eleonora Diella, Cristina Maghini, Chiara Genova, Anna Carla Turconi, Emilia Biffi

**Affiliations:** Scientific Institute, IRCCS Eugenio Medea, 23842 Bosisio Parini, LC, Italy

**Keywords:** navigation skills, cerebral palsy, immersive virtual reality, navigational training, rehabilitation, children

## Abstract

Children with cerebral palsy (CP) suffer deficits in their motor, sensory, and cognitive abilities, as well as in their visuospatial competences. In the last years, several authors have tried to correlate the visuospatial abilities with the navigational ones. Given their importance in everyday functions, navigation skills have been deeply studied using increasingly cutting-edge techniques such as virtual reality (VR). However, to our knowledge, there are no studies focused on training using immersive VR (IVR) in children with movement disorders. For this reason, we proposed an IVR training to 35 young participants with CP and conceived to improve their navigation skills in a “simil-real” environment while playing on a dynamic platform. A subgroup performed a part of the training which was specifically dedicated to the use of the allocentric strategy (i.e., looking for landmarks) to navigate the virtual environment. We then compared the children’s navigation and spatial skills pre- and post-intervention. All the children improved their visual–spatial abilities; particularly, if the IVR activities specifically trained their ability to look for landmarks and use them to navigate. The results of this work highlight the potential of an IVR training program to increase the navigation abilities of patients with CPs.

## 1. Introduction

Cerebral palsy (CP) is a heterogeneous spectrum of non-progressive brain disorders that manifests with motor, sensory, and cognitive deficits [1,2]. Impairments in visual-spatial competences, spatial organization and executive functions are key clinical features in spastic diplegia, a form of bilateral CP where the lower limbs are mainly involved [3,4,5]. This form of CP occurs frequently in preterm children and is generally associated with periventricular leukomalacia (PVL), a white matter injury with neuronal loss and gliosis affecting many projections and associative tracts far beyond those involved in motor planning and control, thus contributing to the complex clinical pattern [6,7,8].

In the last few years, several authors have tried to correlate visual–spatial abilities in patients with CP with their navigational skills [9,10,11,12]. Spatial navigation is the ability to maintain a sense of direction/location while moving in space. In humans, this results in the ability to orientate in complex environments, perceiving distance, and planning routes to distant locations. In addition, it allows individuals to mentally represent the reciprocal relations of landmarks in space [13,14,15]. Navigation skills rely on the integrity of the spatial neural network, which includes the occipital, parietal, frontal, and temporal lobes: the hippocampus and parahippocampal cortex and the retrosplenial and posterior-cingulate cortex (see [16] for a meta-review for fMRI studies). Moreover, several cognitive abilities influence the navigation ones, including long-term memory [17], executive functions [18,19], precision in encoding multisensory experiences, and the ability to form mental representations to guide behavior [20]. Every mobile species needs to navigate the environment to perform functions that involve processes such as perception, learning, memory, and reasoning [13,14,21]. Human navigation abilities largely vary across individuals, change with age, and gradually develop in time [15,22,23]. Several factors influence navigation skills, among which the maturation of sensory and motor systems combined with the environmental feedback, experience-expectant ways, action in the world, and success–failure feedback [15,22,23].

Given its importance, the ability to navigate has been deeply studied over the last few decades to explore the neural mechanisms underlying learning and memory processes [24] and to depict the evolution of spatial skills in human beings throughout their life, especially in early life. In this regard, literature reports that young children learn to navigate in real and virtual environments [25] using two different strategies that can be used alternately or concurrently [26]: the self-based or egocentric strategy (hereafter ES) and the allocentric strategy (hereafter AS). The ES is habit-based and relies on the representations of body parts, implying mainly implicit/procedural memory; the AS is the world-based navigation and it is ground mainly on flexible representations, integrating external objects or environmental features, and depending more on explicit/declarative memory [10,27,28,29,30]. Towards the end of the first decade of life, children reach more sophisticated navigation capabilities and by around 12 years old, we see adult-level performance that consists of integrating and manipulating spatial information [31]. Efficient adult navigation requires both AS and ES strategies and the efficient switch between different reference frames, where flexibility (therefore executive functions) emerges as a determinant of navigation ability [32].

In the past, studies focused on the assessment of navigational skills usually proposed the use of natural and real mazes (see for instance [9,33]), or new electronic devices that translated traditional tests (i.e., Corsi test) to navigational space [10]. However, thanks to the improvement and spread of technologies, virtual reality (VR)-based technologies have progressively replaced them. There are different types of VR-based systems: non-immersive, semi-immersive and immersive VR (IVR). Non-immersive VR systems use small screens (i.e., computer or tablet screens) and the users interact with a mouse or joystick; semi-immersive VR systems have bigger screens and the users can interact with parts of their body; and IVR systems provide immersion and a sense of presence to the users and can be delivered in two major forms: the head-mounted display (HMD) and the computer-assisted virtual environment (Cave). The HMD offers a wide field-of-view stereoscopic vision, while the Cave is a projection-based system whose walls are projection stereo display screens. In this study, we used a Cave system.

Thus, a wide variety of VR tasks has been developed to measure spatial abilities in humans, mostly using non-immersive VR systems [15,30,34,35,36,37,38,39,40]. In these interactive settings, the patient is in front of an ordinary display reproducing a virtual environment, and the individual interacts with the VR environment through a videogame-like controller. Only a few studies propose IVR systems to investigate navigation [12,37,41,42,43], except for the work of Biffi and collaborators [12], and they mainly focus on ageing and dementia. Despite its less common use with respect to non-immersive VR, IVR is an ideal setting to study navigation abilities. Indeed, it allows for the studying of navigation skills in a “simil-real”, ecological, and fully controlled setting, in which the user tends to follow the assumptions about how things work in the real world [44]. In this context, Berthoz and Zaoui [11] suggested several potential paradigms for studying specific deficits in spatial tasks in the developmental age and namely in CP. On the other hand, in 2020, we proposed the assessment of the navigation abilities of young participants with CP [12] using an IVR application which was specifically developed for this scope and based on a 5-way maze located within a playground. We used this application in a pilot study to assess the navigational skills of people with CP and in typically developing (TD) children, who had to navigate and find a reward placed within the maze. What we found was that the ability of children with CP to navigate the space and learn the task was slower than that of TD children, but with a similar final performance. Furthermore, we observed that TD participants mainly relied on allocentric strategies, while egocentric strategies prevailed in participants with CP.

Despite the importance of navigational abilities in everyday life, very little is known about the potential effects of training programs that may help in improving such important skills. Indeed, spatial abilities have proved to be moderately malleable and could improve with specifically designed training programs [45]. In the developmental age, since spatial skills significantly predict educational and occupational achievements in science, technology, engineering, and mathematics (STEM disciplines), early interventions could improve or boost spatial competences (see [46,47] for a spatial training to improve maths skills). Most of the studies that investigate navigation training have focused on improving performance on lower-level spatial functions, such as mental rotation, visual selective attention, spatial problem solving, and landmark recognition [48,49,50,51,52,53,54,55,56]. During the years, researchers have mainly studied the effects of VR navigational training in healthy adults. For example, in 2016, Marusic et al. [57] proposed a computerized navigation training program, with the aim of reducing possible cognitive decline due to bed rest. In 2020, Mc Laren-Gradinaru et al. [58] proposed a computerized training program to improve the ability to form cognitive maps in healthy adult participants, demonstrating its feasibility and effectiveness. The same year, van der Kuil [59] investigated if navigation-preferred strategies in healthy adults (i.e., AS or ES) could change after a VR navigation training, and they found that the training did not affect the objective and self-reported navigation abilities as measured before the training. More recently, Sacco et al. [60] proved that virtual navigation stimulates and enhances the ability to form allocentric representations and spatial memory. To our knowledge, only Boccia et al. [61] investigated the effects of navigational training programs on healthy preschoolers, finding that the training improved their performance on higher-level navigational tasks.

If we know little about the effects of navigational training on healthy subjects, there are even fewer studies aimed at investigating their effects on individuals with pathological conditions. In this context, in 2018, van der Kuil [62] trained the navigation skills of acquired brain injury adult patients that reported navigation impairments using a serious videogame, which resulted in being well appreciated by the participants of the study. A review of 2019 [63] identified sixteen works that explored the effects of navigation training programs on spatial memory in patients with different pathologies (i.e., multiple sclerosis, CP, traumatic brain injury, focal epilepsy, stroke, developmental prosopagnosia, intellectual development disorder, spatial neglect, vertebro–spinal trauma, Alzheimer’s disease, and mild cognitive impairment). Most of these studies focused on adult patients and used both non-immersive [64,65,66,67,68,69,70,71] and semi-immersive [72,73,74,75,76,77] VR-based devices to investigate the assumed changes in navigational skills. The non-immersive training used videogames [65], computer-based applications [64,66,67,68,69,71], and the Khymeia rehabilitation system [70]. On the other hand, the semi-immersive trainings used devices such as the OctaVis [72,73], the BTs Nirvana PC system [75,76], or projectors combined with controllers [74], and/or other objects such as a cycle-ergometer [77]. Despite the previously mentioned studies [64,65,66,67,68,69,70,71,72,73,74,76,77], White and Moussavi [78] treated an adult patient with the probable development of Alzheimer’s disease with an immersive VR apparatus. The device consisted of an Oculus Rift DK2 that ran on a laptop and a customized wheelchair with which the participant could interact with the virtual environment. This study suggested that IVR treatments might benefit people affected by Alzheimer’s disease. Only Akhutina et al. [79] did not perform their study on adult patients, but on subjects with CP in the developmental age range (i.e., 7–14 years old). As to training specifically focused on improving spatial skills in CP, Akhutina et al. examined the effectiveness of a training combining virtual environment instructions with additional desktop tasks, based on the Luria–Vygotsky methodology, for spatial remediation in children having complex motor disabilities restricting their movement. More recently, Di Lieto et al. [80] proposed a computerized training focused on working memory, aimed at improving executive functioning and visuo–spatial skills in children with pre-term spastic diplegia. However, as far as we know, up to now, no specific training has been aimed at specifically improving the navigation skills of children with CP in IVR.

For this reason, here we present a newly developed IVR training program on children, based on looking for landmarks and conceived specifically to improve navigation skills in a “simil-real” environment through a “game-like” training program on a dynamic platform (Motek GRAIL system; see below in Section 2 for details). Indeed, if in 2018 we demonstrated the effectiveness of using this IVR platform in enhancing motor and perceptual competences [81], here we wanted to study the effect of a navigation training specifically thought to be performed on this IVR platform. In particular, the goal of this work was to study navigation and spatial skills pre and post intervention in children with CP who performed an IVR training program and to compare the skills of the group that underwent a landmark-based training for half of the training time with respect to children with CP who did not. We hypothesized that participants with CP trained in the focused navigation group could improve their navigation skills more than their counterparts who trained in IVR without a specific spatial focus. Moreover, a comparison with a control group of TD peers was included to provide a reference of the performance regarding the navigation task.

## 2. Materials and Methods

The design of the study is schematically represented in Figure 1. The participants with CP were recruited and sequentially assigned to the Regular or Navigation IVR training on the GRAIL system (Gait Real-time Analysis Interactive Lab system, by Motek NL). The Regular IVR Training Program aimed at fostering the integration of motor/perceptual competences and improving their walking abilities (Regular Training Program as in Gagliardi et al. [81]). On the other side, the IVR Navigation Training Program had an identical duration/frequency and consisted in a Regular IVR training on the GRAIL platform, enhanced with a navigation training based on looking for landmarks (see Section 2.4 for details). The participants with CP underwent two assessments, at the beginning of the training program (T0) and after 4 weeks (T1). In particular, we assessed the navigational skills and visual–spatial abilities at T0 and T1 (see Section 2.2 and Section 2.3 for details), while the cognitive competences were measured only at T0, using Raven’s progressive matrices [82], as in Biffi et al. [12]. During the interposed 4 weeks, they performed 18 daily IVR training sessions, according to the program appropriate for the membership group. We also proposed the assessments at T0 and T1 to a group of typically developing (TD) children, with the same time gap, in order to have a reference performance regarding the navigation task.

Details about the inclusion criteria, the evaluation of the spatial and motor abilities, and the training programs are deeply described below.

The Ethics Committee of the Scientific Institute approved the study protocol. Written informed consent to participate in this study was provided by each participants’ legal guardian/next of kin in accordance with the Declaration of Helsinki. The study has been registered as a clinical trial on ClinicalTrials.gov (NCT04270305).

### 2.1. Participants

Patients were recruited in the Neuro-rehabilitation Unit of the Scientific Institute IRCCS E. Medea in Italy. The inclusion criteria were: a diagnosis of bilateral CP; aged between 7 and 15 years old; the severity of their motor impairment classified as I, II, and III, according to the Gross Motor Function Classification System (GMFCS) [83], and to the Manual Ability Classification System (MACS) [84]; and the ability to follow the instructions. The exclusion criteria were as follows: severe muscle spasticity and/or contracture, a diagnosis of a severe learning disability, behavioral problems, and visual difficulties that would affect the proposed activity and participation (i.e., Snellen Visual Acuity <3/10”).

### 2.2. Evaluation of the Visuospatial Abilities

Spatial abilities were assessed in all participants both after the recruitment (T0) and after the 4-weeks training sessions (T1) by two classical paper-and-pencil tests: the Corsi Block Test [85] and the Labyrinth subtest of the WISC-III (Wechsler Intelligence Scale for Children, third) [86]. The Corsi Block Test evaluates visuospatial memory and consists of a set of identical wooden blocks arranged in a way that they were not aligned on a desktop. The tester taps on a number of blocks in sequence, and the participant is asked to reproduce the tapped sequence, which is of variable lengths. The Labyrinth subtest consists of 10 paper-and-pencil tasks with an increasing complexity and measures one’s planning ability, perceptual organization, visual–motor coordination, and self-control. During the test, the participant has to find their way out from the center of a two-dimensional maze.

### 2.3. Evaluation of the Navigation Abilities in IVR

Navigation skills were assessed at T0 and T1, as described in Biffi et al. [12], using an IVR application, the “Star-Maze” app (described below) on the GRAIL system, which integrates IVR environments projected on a 180° cylindrical screen, a Vicon motion-capture system (Oxford Metrics, Oxford, UK), and a two-degree of freedom platform. Each participant had to navigate in a 5-way maze in a playground (Figure 2) to find a reward. To turn right or left within the IVR scene, the subject had to shift the pelvis right or left. Furthermore, the IVR environment accelerated or decelerated when he/she moved forward or backward. The interactive scenario included five alleys radiated from the angles of a regular pentagon in the center of the maze; each alley is characterized by environmental cues (e.g., swings, slides, houses, and mountains) that make it similar but different from the others (see [12] for details about the maze design).

For the evaluation of the navigational abilities, two different tasks were proposed: the Free Navigation and the Compelled Strategies Navigation Tasks.

In the Free Navigation Task (Figure 3), the participant had to search for the treasure 21 times. In sixteen attempts (hereinafter training trials), the participant started from alley 1 and could freely move in the maze, looking for the treasure that was located in alley 3 (according to [12,34]) (Figure 3b). The subject had two minutes to find the treasure; if he/she reached alley 3 in this time, he/she succeeded. Five testing trials were interposed (see Figure 3a) to assess whether he/she resorted to the AS or ES to solve the task. In these 5 trials, the participant started the navigation in alley 4 and the treasure was in alley 3 or in alley 1 (Figure 3c). Particularly, if he/she recognized environmental cues and entered Alley 3, the participant relied on the AS. On the other hand, if he/she used the same trajectory performed during the training trials and thus reached alley 1, he/she used the ES (Figure 3c). The variations of the conditions in the five interposed trials were never explicitly disclosed to the participant. Each navigation lasted up to 2 min, with a stable verbal instruction “look for the treasure”.

In the Compelled Strategies Navigation Task (Figure 3d), the participant was asked to resort to a given strategy to get the treasure, as previously done also in [34]. During the “compelled AS” trials, the participant started the navigation in a new alley (alley 2 or 5) and the treasure was in alley 3 (Figure 3e). Therefore, the treasure could be reached only by integrating environmental cues (i.e., the AS). According to Igloi et al. [34], the participants had to perform 4 “compelled AS” trials. During the “compelled ES” trials, the participant started in alley 1 and the treasure was located in alley 3, as done in the training trials, but the landmarks were removed (Figure 3f) and only the five alleys, the grass, and the surrounding mountains were visible. This compelled the participant to resort to body/route-based strategies. The sequence of the trials (the imposed AS and imposed ES) was counterbalanced.

### 2.4. Regular and Navigation Training Programs in IVR

As previously reported, following the assessment at T0, both the CP-Navigation and CP-Regular group underwent a 4-week IVR training program that ended at T1. The 16 subjects belonging to the CP-Navigation group performed the IVR Navigation Training Program, while the 11 subjects belonging to the CP-Regular group underwent the IVR regular one. Both the trainings were performed on the GRAIL system, which provides a dual-belt treadmill, a two degrees of freedom motion platform (10° of pitch, 5 cm of sway), a Vicon motion-capture system, and a 180° cylindrical screen. The Vicon system allows for conducting a motion analysis by means of spherical 15 mm markers placed on specific body landmarks. The system is a medical device dedicated to motor and cognitive rehabilitation.

#### 2.4.1. The Regular IVR Training Program

The Regular Training Program aimed at improving walking and balance abilities using engaging VR environments (training program as in Gagliardi et al. [81]). The program consisted of 18 45 min training sessions, during which the operator proposed five different games which were available in the Motek application package of the GRAIL system, aimed at improving the different aspects of the children’s motor difficulties. The program was designed during the clinical assessment (T0) according to the motor needs of each child with CP, during both standing and walking conditions. The standing activities aimed to improve the body weight transfer between the two legs and the static balance or stranding reaction. In addition, the standing tasks allowed patients with CP to train their single limb support and better control their center of mass, shifting it from the standing limb to the trailing. On the other hand, the walking activities aimed to gradually increase the walking speed and resistance or improve the balance of the patients, even in irregular ground. Several walking activities were intended to be multitasking, and the subject had to walk while catching objects with markers on their hands, or to deal with cognitive tasks (i.e., Stroop and calculus). When clinical conditions allowed it, we proposed activities directed to improve the patient’s gait pattern.

#### 2.4.2. The IVR Navigation Training Program

Four applications were specifically used on the GRAIL system to train the navigational abilities and the visuospatial organization skills: the Water labyrinth, the Grass labyrinth, the Dinosaurs, and the Boat. These applications aimed at improving the online processing of landmarks during the navigation, both during the free exploration of the environments (such as in the Dinosaurs and Boats apps), and with the presence of spatial constraints (such as in the Water and Grass Labyrinths) (see below for details).

While the Boat scenario was already present in the Motek application package of the GRAIL system and was only modified, the other three scenarios were modelled using Google SketchUp and imported into Autodesk3ds Max as Collada files. Then, the scene was exported into Ogre format and uploaded in the D-flow software. Some objects from the Motek application package (e.g., animals) were imported into the two labyrinths and Dinosaurs scenarios. The Boat scenario was modified by removing some useless scene objects and creating new solid geometrical figures with different colors and shapes that were placed into the environment. The objects inserted within the scenarios acted as landmarks in the two labyrinths, and as targets to catch in the Dinosaurs and Boat applications. Different difficulty levels were implemented for each application through the D-flow software, to make the training more engaging. The participant could navigate within these environments by the means of the movement of his/her pelvis, as traced by two passive reflective markers. In particular, forward/backward pelvis movements were converted into the subject’s acceleration/deceleration, while a left/right pelvic shift was converted into left/right turning.

The Navigation Training Program consisted of 18 45 min sessions, during which the participant tested each of the four different applications for 5 min, and the remaining time was dedicated to regular walking exercises on the GRAIL system (see the paragraph named “The Regular IVR Training Program” for details about these exercises). During the training, the therapist increased the difficulty level, starting from the easiest according to the subject’s skills.

In the Dinosaurs application, the patient could move autonomously in the virtual environment, represented by a park with dinosaurs of various types and sizes. The aim was to rescue the animals placed into the virtual environment and lead them to a waterfall (Figure 4a). The movement in the virtual environment is guided by two markers placed on the participant’s lower back. When the subject reached the animal, it was “rescue”. When the participants rescued all the animals, he/she led them to the waterfall with his/her movement. The sequence of animals to be taken to the waterfall was set a priori to choosing the level. In the Boat application, the subject was on a virtual boat and had to capture the geometrical figures present in the scenario, following the sequence set by the chosen level. The goal was to arrive with these objects at the finishing line (Figure 4c). These two applications, characterized by an open space without a pre-set path, aimed at training the subject to use visual–spatial organization skills. Indeed, they required patients to reach and take the objects in a sequence, leaving them free to move within the virtual environment.

In the Grass and Water labyrinth applications, the subject was free to explore and move around in the virtual environment, consisting of a maze made entirely of grass or water (Figure 4b,d). The aim of the task was to find the exit in the shortest possible time. The landmarks (balloon animals) were placed inside the labyrinth in order to allow the subject to orient him/herself; the number of landmarks decreased while the level of difficulty was increasing, with the highest level being without any landmarks to compel the participants to use an ES for reaching the goal. The operator could choose the level of the game, thus defining the complexity and the landmarks (if any) to be shown during the task. These two applications were specifically dedicated to the training of the individual’s navigational abilities by prompting subjects first to focus on landmarks and then to learn the path.

### 2.5. Data Analysis and Statistics

All the statistical analyses were carried out with IBM SPSS Statistics v21, setting the significance level at 5%.

#### 2.5.1. Demographic Data and Visuospatial Abilities

Descriptive results were obtained for each participant and for all the variables (see Tables 1 and 2) for the pre/post-intervention outcome. In order to compare among different ages and tests, the results are expressed for each participant as Z-scores. In particular, Z-scores were computed for Raven’s Colored Progressive Matrices (Raven Z-score hereinafter), the Corsi Block Test (Corsi Z-score hereinafter), and the Labyrinth Test from WISC-III (Labyrinth Z-score hereinafter).

First, we performed a chi-square test on the demographic data and Z-scores of all the participants with CP to verify the uniformity of their distribution before the partition into two subgroups. Then, the demographic data and visuospatial abilities were compared among CP-Navigation, CP-Regular, and TD groups with non-parametric tests. Specifically, the number of males and females, the GMFCS, and the MACS levels were compared by means of the chi-square test for uniformity, while the participants’ age, the Raven, the Corsi, and the Labyrinth Z-scores with the Kruskal–Wallis test, and the Bonferroni corrected Mann–Whitney test as a post hoc analysis. The Wilcoxon test was used to evaluate if the Corsi and Labyrinth Z-scores changed from T0 to T1 in each group. Finally, we computed the Corsi and Labyrinth delta Z-scores between T0 and T1, analyzing if there were statistically significant differences among the three groups with the Kruskal–Wallis test.

The participant’s age is reported in Section 3 as the mean (standard deviation-SD), while not normally distributed data are shown as the median (interquartile range-IQR).

#### 2.5.2. Star-Maze Scoring

During the navigation, the “Star-Maze” application recorded the participant’s position in the virtual environment and his/her movements within a Cartesian coordinate system. The success or lack of success in finding the treasure, the number of visited alleys to get the treasure, and the total path length (TPL) travelled to find the treasure were computed for each trial with a custom-made software developed in Matlab (Mathworks).

As formerly described in [12], the distance error (DE, Equation (1)) and the rotation angle (RA, Equation (2)), which assesses the number of rotations, were then computed to qualify the efficiency of the navigation skills (the higher the value, the worse the performance in both parameters).
(1)$$\mathrm{DE}\;\left(\%\right)=100\;*\;\frac{\mathrm{total}\;\mathrm{distance}\;\mathrm{travelled}-\mathrm{ideal}\;\mathrm{distance}\;\mathrm{travelled}}{\mathrm{ideal}\;\mathrm{distance}\;\mathrm{travelled}}$$
(2)$$\mathrm{RA}\;\left(\mathrm{degrees}\right)=\mathrm{Participant}’s\;\mathrm{rotations}-\mathrm{minimum}\;\mathrm{rotations}$$

These parameters were computed in all the 16 trials of the “Free Navigation Task”. Then, we assessed how many trials were required to stably find the treasure, i.e., to succeed in learning the way towards the treasure (success/no success parameter). In addition, the number of visited alleys, TPL, DE, and RA within the 16 trials allowed for following the learning process in the navigation of an IVR maze for each participant. In our previous work [12], we identified the stabilization trials (i.e., the trial after which the performance is stable) as the 5th and the 7th ones, for the TDs and the CPs, respectively. To verify that the performance was stable after these trials on this new dataset, the Friedman test was performed on the number of visited alleys, TPL, DE, and RA. Then, the navigation performances at the “stabilization trial” were compared among TDs, CP-Navigation, and CP-Regular by means of a Kruskal–Wallis test, since we aimed at assessing the potential differences after the learning phase. On the other hand, the Wilcoxon test was used to evaluate, for each group, if the success rate statistically changed from T0 to T1. Finally, we computed the delta of the Star Maze scores (the number of visited alleys, TPL, DE, and RA) between T0 and T1, analyzing if there were statistically significant differences among the three groups with the Kruskal–Wallis test.

Each testing trial of the Free Navigation Task was named as AS if the participant used the allocentric strategy to solve the task, with ES if he/she used the egocentric strategy, or with no strategy if he/she did not succeed. Then, the overall strategy freely adopted by the participants was determined as in [12], and the subjects were classified in the allocentric, egocentric, shifter, or lacking in efficient strategy (no-strategy hereafter). In particular, allocentric and egocentric individuals are those who used, respectively, an AS or ES in more than three consecutive testing trials, while shifters are those who use both the ES and AS along with the trials. No-strategy individuals were those participants who did not succeed in more than two testing trials.

During the Compelled Strategies Navigation Task, successful trials were those where the participant found the treasure following a direct route from the starting point to the alley with the reward. For each participant, the percentage of success in reaching the reward was calculated separately for the imposed allocentric and egocentric navigations. Afterwards, the median values (and IQR ranges) for the whole group and for the ES and AS subgroups (as defined in the Free Navigation Task) were computed for the imposed AS and ES. We compared the performance among CP-Navigation, CP-Regular, and TD groups with the Kruskal–Wallis test, while the Wilcoxon test was used to evaluate, for each group, if the success rate statistically changed from T0 to T1. Additionally, we computed the delta of the performance scoring between T0 and T1, analyzing if there were statistically significant differences among the three groups with the Kruskal–Wallis test.

## 3. Results

### 3.1. Demographic Data and Visuospatial Abilities

According to the inclusion and exclusion criteria, 35 children suffering from bilateral CP were recruited, while 3 were excluded. Thus, 32 participants were sequentially assigned to the Navigation and to the Regular IVR training group. After the assignment, five more participants declined to participate. The study thus included 27 participants with CP (21 males, 6 females, a mean age of 10.6 ± 1.3 years old). The group of CP children was uniformly distributed in terms of their age, GMFCS classification, and Raven Z-scores (minimum value of *p* equal to 0.097), but not in terms of their gender and the MACS classification (*p*_gender_ = 0.004; *p*_MACS_ = 0.008). The allocation within the Navigation and Regular training group was sequential. The navigation training group consisted of 16 participants (CP-Navigation group hereafter, 13 males, 3 females, a mean age of 11.5 ± 2.2 years old) classified as to GMFCS I/II/III: 9/0/7 and as to MACS I/II/III: 9/5/2. The Regular group included 11 participants (CP-Regular group hereafter, 8 males, 3 females, a mean age of 9.6 ± 1.7 years old) classified as to GMFCS I/II/III: 4/4/3 and as to MACS I/II/III: 7/3/1. A control group of TD peers was included to provide a reference of the performance at the navigation task at the given times. Fourteen TD participants were recruited (6 males, 8 females, a mean age of 10.7 ± 2.6 years old). All TD participants were healthy, with no history of a psychiatric or neurological illness, learning disabilities, and hearing or visual loss; they showed average school performances in language, arts, and reading. Table 1 shows the demographic details of the children, as well as the median and interquartile range of the Raven Z-scores.

**Table 1 jcm-11-06146-t001:** Demographic features of participants.

Demographic Feature	TDs	CPs-Regular	CPs-Navigation	*p*
Gender M/F	6/8	8/3	13/3	0.074 ^&^
Age (years) ^$^	10.7 ± 2.6	9.6 ± 1.7	11.5 ± 2.2	0.081 ^&&^
GMFCS I/II/III	-	4/4/3	9/0/7	**0.033** ^&^
MACS I/II/III	-	7/3/1	9/5/2	0.922 ^&^
Raven’s Colored Progressive Matrices ^£^	1.63 (1.45)	0.00 (0.80) *	−0.07 (2.64) *	**0.033** ^&&^

M, males and F, females; significant values of *p* are in bold. ^$^ Mean and standard deviation. ^£^ Median and interquartile range. ^&^ Value of *p* of the chi-square test for uniformity. ^&&^ Value of *p* of the Kruskal–Wallis test. * Post hoc analysis shows a statistical different distribution of the Raven Z-score between CPs-Regular and TDs and between CPs-Navigation and TDs. Values of *p* in the main text.

The three groups were comparable as to their age and gender, while, not surprisingly, the participants with CP performed significantly worse than TDs at the Raven’s Colored Progressive Matrices [Raven Z-score: CP-Navigation −0.07 (2.64), CP-Regular 0 (0.8), and TD 1.63 (1.45), *p* = 0.003]. In detail, the post hoc analysis showed significant differences between CPs-Navigation and TDs (*p* = 0.004) and CPs-Regular and TDs (*p* = 0.030), while no significant difference between CPs-Regular and CPs-Navigation (*p* = 1.000).

Comparing the GMFCS, the CP-Regular and CP-Navigation children’s scores resulted in being statistically different (*p =* 0.033). Indeed, most of the CPs-Navigation had a low impairment of the gross motor functions (56% with level I), while most of the CPs-Regular had a higher level of impairment (64% with level II or III).

Regarding the visuospatial abilities, we found a uniform distribution of the Corsi and Labyrinth Z-scores when considering the whole group of CP participants (*p*_Corsi_ = 1.000; *p*_Labyrinth_ = 0.859). The results investigating the visuospatial abilities when considering the two CP subgroups and the TDs are reported in Table 2. As expected, TD participants performed better than children with CP at T0 (Kruskal–Wallis test, Corsi Z-score: *p*T0 = 0.032; Labyrinth Z-score: *p*T0 = 0.001). The post hoc analysis showed a significant difference in Labyrinth Z-scores both between CPs-Navigation and TDs (*p* = 0.004) and between CPs-Regular and TDs (*p* = 0.004), while the Corsi Z-scores were significantly different only between CPs-Navigation and TDs (*p* = 0.037). No statistical difference was found among CPs-Regular and TDs (*p* = 0.169), suggesting that the recruited CPs-Regular had a better visuospatial memory than CPs-Navigation at the baseline.

Interestingly, the Kruskal–Wallis test at T1 showed no statistically significant differences (Corsi Z-score: *p*T1 = 0.830; Labyrinth Z-score: *p*T1 = 0.052), suggesting an improvement in the Labyrinth for both groups and in the Corsi test for the CP-Navigation group.

Comparing participants’ performance between T0 and T1, Table 2 shows a post-training improvement of the visuospatial abilities in children with CP from T0 to T1. However, this improvement was statistically significant as to the Labyrinth subtest with only a trend in the CP-Navigation group for the Corsi block test (Labyrinth Z-score: *p*CP-Navigation = 0.035, *p*CP-Regular = 0.018, *p*TD = 0.6738; Corsi Z-score: *p*CP-Navigation = 0.074, *p*CP-Regular = 0.581, *p*TD = 0.068). No significant improvement was detected in the TDs group. Finally, the Corsi and Labyrinth delta Z-scores between T0 and T1 did not show statistically significant differences among the groups, even if the Labyrinth delta Z-score showed a *p*-value close to 0.05 (*p*_ΔCorsi_ = 0.830, *p*_ΔLabyrinth_ = 0.056).

**Table 2 jcm-11-06146-t002:** Visual spatial competences in CP and TD participants at T0 and T1.

Test		TDs	CPs-Regular	CPs-Navigation	*p* ^&^
Corsi block test Z-score	T0	0.350 (1.879)	0.125 (2.360)	−0.590 (1.845) *	**0.032**
	T1	0.725 (1.594)	0.130 (2.360)	−0.4150 (2.325)	0.830
	*p* ^$^	0.068	0.581	0.074	-
Labyrinth subtest Z-score	T0	1.165 (1.250)	−1.660 (2.130) °	−1.500 (3.230) °	**0.001**
	T1	1.315 (1.498)	−1.330 (2.660)	−0.165 (2.900)	0.056
	*p* ^$^	0.673	**0.018**	**0.035**	-

Median values and interquartile ranges are shown; significant values of *p* are in bold. ^&^ Value of *p* of the Kruskal–Wallis test. Null hypothesis: Corsi and Labyrinth Z-scores distribution at T0 (or T1) is the same for CP-Navigation, CP-Regular, and TD groups. * Post hoc analysis shows a statistical different distribution of the Corsi Z-score between CPs-Navigation and TDs at T0. Values of *p* in the main text. ° Post hoc analysis shows a statistical different distribution of the Labyrinth Z-score between CPs-Regular and TDs and between CPs-Navigation and TDs (T0). Values of *p* in the main text. ^$^ Value of *p* of the Wilcoxon test. Null hypothesis: Corsi and Labyrinth Z-scores distribution for CP-Navigation, CP-Regular, and TD groups is the same at T0 and T1.

### 3.2. Star-Maze Scoring during the Free Navigational Task

Considering the Free Navigational Task at T0, six TDs (43%) succeeded in learning the way through the virtual star maze from the first trial and never failed subsequently, six (43%) within the second/third trial, while two participants (14%) took at least 4 trials to stably succeed. On the other side, seven of the CP-Navigation children (44%) learned the maze from the first trial, while four of them (25%) had to perform 2/3 trials before they succeeded, four (25%) at least 4/7 trials, and one patient (6%) never succeeded. Finally, only two CP-Regular children (20%) managed to find the treasure at the first trial, four of them (40%) at the second/third one, four (40%) at the fourth/seventh one, and one patient (10%) never succeeded.

The same success/no success parameter was calculated for all the three groups at T1. All the TDs succeeded in finding the treasure from the first trial, even if they did not have any type of training. Even the children with CP succeeded in finding the treasure from the first trial, with the exception of one participant of the CP-Regular group, who managed to find the treasure at the fourth trial.

Beyond calculating the success/no success parameter, we investigated the navigational performance in the virtual maze, before (T0) and after (T1) the Navigation or Regular training program. Figure 5 shows the learning curves in terms of the number of visited alleys, TPL, DE, and RA of the three groups (TDs in orange, CPs-Regular in blue, and CPs-Navigational in grey), on the left referred to T0, while on the right to T1. The Friedman test for the trials after the 5th and 7th ones, for all the learning parameters (i.e., the number of visited alleys, TPL, DE and RA) confirmed the previous findings [12] with no statistical differences for all the learning parameters, for the TDs, the CPs-Regular, and the CPs-Navigation (minimum *p*-values for the Friedman test: TDs = 0.386, CPs-Regular = 0.235, CPs-Navigation = 0.147). Therefore, the results at T0 show that children with CP took more trials to stabilize the characteristics of the performance. However, as reported in Table 3, all the three groups did not differ significantly when stable for any of the learning parameters (minimum value of *p* for the Kruskal–Wallis test = 0.087). The only exception refers to the RA (*p*T0 = 0.034), for which the post hoc analysis showed a significant difference between TDs and CPs-Regular (*p* = 0.029), having the latter higher values when stable.

Unlike the curves referred to at T0, the ones obtained at T1 were almost stabilized for any of the learning parameters, meaning that all the participants (both TDs and CPs) had a stable performance from the very beginning, and the learning process did not occur anymore. In particular, when performing the Friedman test for all the learning parameters (number of visited alleys, TPL, DE and RA), we identified that the performance was stable from the second trial for all the three groups (minimum *p*-value for the Friedman test as follows: *p*TDs = 0.308, *p*CP-Regular = 0.149 and *p*CP-Navigation = 0.161). In addition, Table 3 shows that, at T1, the performance of the three groups at the stable trial (the 2^nd^ one) were comparable (minimum value of *p* for the Kruskal–Wallis test = 0.268), with the exception of the RA (*p* = 0.019), for which we found a significant difference between TDs and CPs-Regular (*p*-value = 0.032), as at T0.

Finally, Table 3 shows no statistically significant differences in the performances at T0 and T1 once the stabilization trial had been reached (TDs: fifth trial at T0, second trial at T1; CPs: seventh trial at T0, second trial at T1), with the exception of the TPL parameter for the CP-Navigation participants (*p* = 0.044), which optimized their navigation at T1. The delta of the performance scores between T0 and T1 did not show statistically significant differences among the groups (*p*_ΔVisitedAlley_ = 0.520, *p*_ΔDE_ = 0.520, *p*_ΔTPL_ = 0.210, *p*_ΔRA_ = 0.693).

Table 4 reports the strategy freely adopted by each participant during the five testing trials, both at T0 and T1. The results show that all the TDs adopted a strategy to complete the task already at T0: the majority of TD children adopted the AS (50%), 36% the ES, while 14% were shifters. On the other side, both the CP groups performed worse than the TDs, as expected, since some subjects did not adopt an efficient strategy at T0 (9% of the CP-Regular and 19% of the CP-Navigation children). However, children with CP improved their abilities at T1. In particular, all the CPs adopted a strategy to complete the task (i.e., none of the CPs was classified as “No strategy” at T1). In addition, most of the CP-Navigation participants started to adopt the AS, while the CP-Regular group increased the number of allocentric subjects, but not reaching the majority.

### 3.3. Star-Maze Scoring during the Compelled Strategies Navigation Task

Finally, during the Compelled Strategies Navigation Task, the participants were asked to modify their strategies, adopting either an imposed AS or ES. Table 5 reports the success scores at T0 and T1, both considering the imposed AS and ES tasks.

In general, TDs had a more stable performance than CPs at both T0 and T1. No statistically significant difference among the success scores was found among the groups, with the exception of the scores at T1 during the imposed ES (*p* = 0.04); however, the post hoc analysis did not show any significant differences between the paired groups. As expected, TDs did not show any significant improvement from T0 to T1 during neither the imposed AS (*p* = 0.739) nor the imposed ES (*p* = 0.416). Regarding the children with CP, during the imposed AS trial, CPs-Navigation statistically improved the success rate from T0 to T1 (*p* = 0.018), while CPs-Regular had a non-significantly worse performance at T1 rather than T0 (*p* = 0.157). On the other side, during the imposed ES trial, CPs-Navigation performed non-significantly worse at T1 than T0 (*p* = 0.310), while CPs-Regular slightly improved their abilities from T0 to T1 (*p* = 0.227). The delta of the success scores between T0 and T1 did not show statistically significant differences among the three groups (*p*_imposed-AS_ = 0.178, *p*_imposed-ES_ = 0.102).

Finally, Figure 6 shows the percentage of success for the trials with the imposed AS and the imposed ES, both at T0 and T1, in TD, CP-Regular, and CP-Navigation participants who were grouped according to their orienting strategy that was adopted during the Free Navigation Task (AS and ES subgroups). From this figure emerges that participants tended to perform better when the imposed strategy coincided with the one they mostly chose during the Free Navigation Task: AS children performed better during the imposed AS task, while ES during the imposed ES one. Furthermore, Figure 6 seems to suggest that the performance improved at T1.

## 4. Discussion

The role of navigation skills is fundamental to perform everyday functions and, for this reason, over the past decades, researchers have deeply studied the ability to navigate. What emerged is that navigation skills largely vary across individuals, change with age, and gradually develop in time. In particular, young children use the ES or the AS when navigating in the environment; only at around 12 years old do their capabilities become more sophisticated, reaching an adult-level performance that consists of flexibly switching from the AS to ES and vice versa, depending on the circumstances.

However, children with impairments in visual–spatial competences may experience difficulties in evolving their navigational skills. This is the case for children with CP that manifest motor, sensory, and cognitive deficits, as well as impairments in visual–spatial competences, spatial organization, and executive functions.

Immersive VR can be an ideal setting to study navigational abilities in patients with CP, since it consists of a “simil-real”, ecological, and fully controlled setting. This was demonstrated by our previous study in 2020, which showed the effectiveness of using an IVR dynamic platform in studying the navigation learning process during an IVR task and verifying the efficacy and flexibility of navigation after the learning [12]. Therefore, here we wanted to study the effect of navigation training on this IVR platform, comparing navigation and spatial skills pre- and post-intervention among children with CP. In this study, we included children with CP that performed a training in IVR and a subgroup that performed a part of training specifically dedicated to navigation by using landmarks. Furthermore, we enrolled TD peers to provide a reference of the performance regarding the navigation task. The training had a game-like imprint and consisted of looking for landmarks in simil-real IVR environments.

What we found is an improvement in the visuospatial abilities of all children with CP after both the trainings, which highlights the efficiency of IVR programs. However, since this improvement was statistically significant only regarding the Labyrinth subtest and not the Corsi block test, this suggests that the improvement mainly concerned the planning ability, perceptual organization, visual–motor coordination, and self-control, rather than the visuospatial memory.

Regarding the Star-Maze scoring, the curves in Figure 5 referred to the learning parameters (i.e., the number of visited alleys, TPL, DE, and RA) demonstrated that all participants (CPs and TDs) progressively learnt their ways through the maze at T0. This learning was maintained and present at T1. Indeed, at T0 all participants succeeded in finding the treasure and stabilizing their performance after some trials, while at T1 they already succeeded at the first one, as demonstrated by the flat learning curves on the right side of Figure 5. The learning parameters at the stable trial did not differ significantly for all the three groups at both T0 and T1, with the exception of the RA, for which the post hoc analysis showed a significant difference between TDs and CPs-Regular. However, while at T0 the CPs reached the stable trial at the seventh attempt, the TDs were faster and had a stable performance already at the fifth one. On the contrary, at T1, the stable trial corresponded to the second one for both the TDs and the CPs, confirming that the learning process of the task occurred mainly at T0 and its stability after one month. The significant difference between the TDs and CPs-Regular regarding the RA at T0 and T1 might mean that the CPs-Regular had worse navigation abilities than the other groups and that they did not level their navigation skills to the other groups after the regular training program.

Considering the strategies adopted during the trials (i.e., an AS, ES, a shifter or without strategy), we found that both CP groups modified their strategies from T0 to T1 and, at T1, any of the CPs were no more classified as “without strategy”. In addition, at T1, most of the CP-Navigation participants started to adopt the AS, while the CP-Regular group increased the number of allocentric subjects, but not reaching the majority. Therefore, the IVR training improved the navigation strategies in all participants, who learnt to navigate more efficiently though the maze; only those participants who trained their navigational skills improved their online processing of landmarks. Thanks to this learning, by the end of the training the CP-Navigation participants could rely on the flexible representations that depend on explicit/declarative memory and use an AS, which is the one that can be trained despite the ES.

The success scores obtained during the Compelled Strategies Navigation Task highlight that the navigation training program led to better results in navigation skills than the regular training during the imposed AS trials, since the training was based on landmarks. Indeed, despite the CPs-Regular, the CPs-Navigation statistically improved the success rate from T0 to T1. It is important to underline that, after the 4-weeks trainings, most of the CPs became AS oriented (from 19% at T0 to 56% at T1), while the number of children who freely adopted the ES during the Free Navigation Task decreased from 52% to 30% (see also Table 4 for more details). This means that participants tended to perform better when the imposed strategy coincided with their orienting one which was adopted during the Free Navigation Task. Therefore, if considering that the number of AS-oriented subjects became higher at T1, this means that the success scores at T1 during the imposed AS trials are more relevant than the ones obtained during the imposed ES since they mediated among more participants.

In general, we found that all the children with CP improved their performance and the motor efficiency after the IVR trainings, thanks to the motor activity performed in a fully controlled setting. Indeed, during the trainings, the patients learned “how to move” and “where to go” on the GRAIL platform and started to navigate more efficiently, improving their ability of changing direction, processing simultaneously the stimuli and integrating the landmarks. However, the group that dedicated part of the session to the navigation training had an improvement in the capacity to integrate the landmarks, as demonstrated by the higher number of subjects that started to use an AS.

The CP-Regular group also improved the performance, but the absence of a training specifically aimed at integrating the space references led to more random improvements, given by time and self-organization. However, even if the navigation training improved the ability to identify “where to go” in routine conditions, thanks to the transition to the AS and the landmark integration, it was not sufficient to reach an efficient navigation during the compelled conditions. This is probably due to the training time that was not sufficient to allow the consolidation of what they learned.

This work has some limitations. The first one is that, even if all the children with CP were uniformly distributed in terms of GMFCS, Corsi, and Labyrinth scores at the recruitment, after the random subdivision into the Regular and Navigation groups, the motor impairment (i.e., GMFCS scores) resulted in being statistically different between the groups given the smallness of the resulting groups, and this could potentially influence the result. A second limitation of this work is that the duration of the navigation training was short: only 20 min out of the 45 of the entire training were dedicated to the navigation program, while 25 min to the regular motor program. Nevertheless, participants with CP who performed the training whilst being more focused on the navigational abilities learnt to integrate landmarks and to use the AS beyond the ES. Another limitation is that cybersickness, a symptom of motion sickness often experienced in VR and IVR systems, was not evaluated using specific questionnaires. Nevertheless, during the trainings, the therapists always asked the children if they were experiencing any type of sickness, and they did not report any side effects. Finally, the GRAIL system we used for the IVR training is a half-open environment that is less realistic than devices able to provide a totally immersive environment, and the scenario was graphically poor in details. For this reason, our future work will aim at investigating head-mounted displays, evaluating the cybersickness, and making the scenarios more realistic.

## 5. Conclusions

To conclude, the results of this work highlight the potential of an IVR training program to increase the navigational abilities in patients with CPs. Particularly, if the IVR activities specifically train the ability to look for landmarks and to use them to navigate, the results in terms of the use of an allocentric strategy are even better. Future studies will test a full-time focused navigation training to assess its potential and to integrate it in the whole process of the rehabilitation of children with CP, assessing also the transferability and the relapse of the training on everyday life.

## Figures and Tables

**Figure 1 jcm-11-06146-f001:**
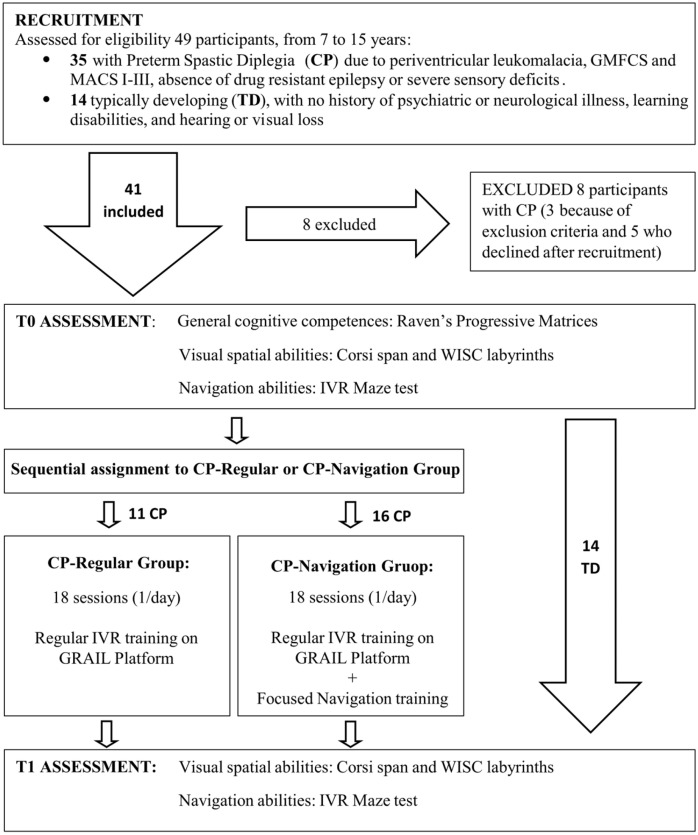
Design of the study.

**Figure 2 jcm-11-06146-f002:**
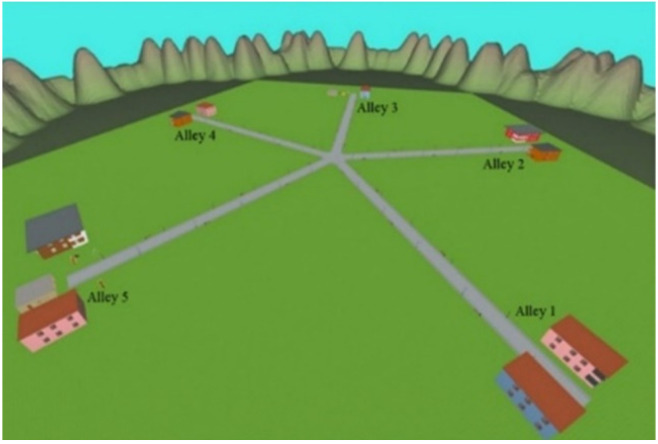
Immersive Virtual Reality scenario: the 5-way maze used for evaluating the navigational skills at T0 and T1.

**Figure 3 jcm-11-06146-f003:**
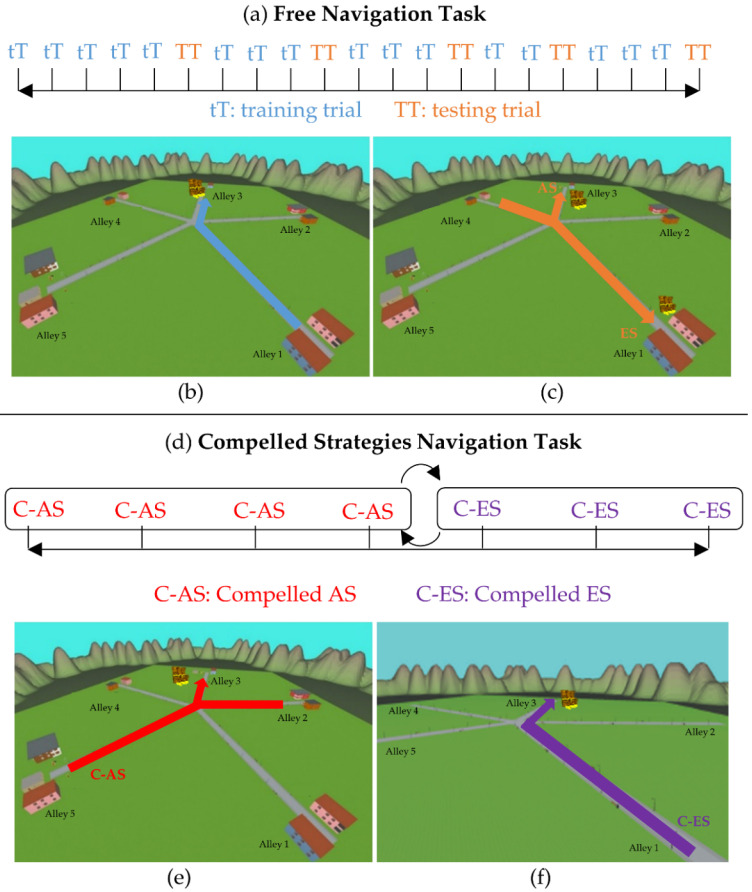
(**a**) Schematic representation of the Free Navigation Task in which the participant had to search for the treasure 21 times: 16 trials of training and 5 trials interposed to assess whether he/she resorted to AS or ES to solve the task. (**b**) Path to be followed during the training trials in the Free Navigation Task (blue path). (**c**) Path to be followed during the testing trials in the Free Navigation Task (orange path): the subject resort to an allocentric strategy if he/she goes to alley 3, to the egocentric if he/she goes to alley 1; (**d**) schematic representation of the Compelled Strategies Navigation Task in which the participant was asked to resort to a given strategy to get the treasure: there were four “compelled AS” trials and three “compelled ES” trials; the order of the egocentric and the allocentric versions of the task were counterbalanced among the subjects. (**e**) Two paths imposed during compelled allocentric trials (red paths). (**f**) The bare maze used in the Compelled Strategies Navigation Task with the path imposed during the compelled ES (violet path).

**Figure 4 jcm-11-06146-f004:**
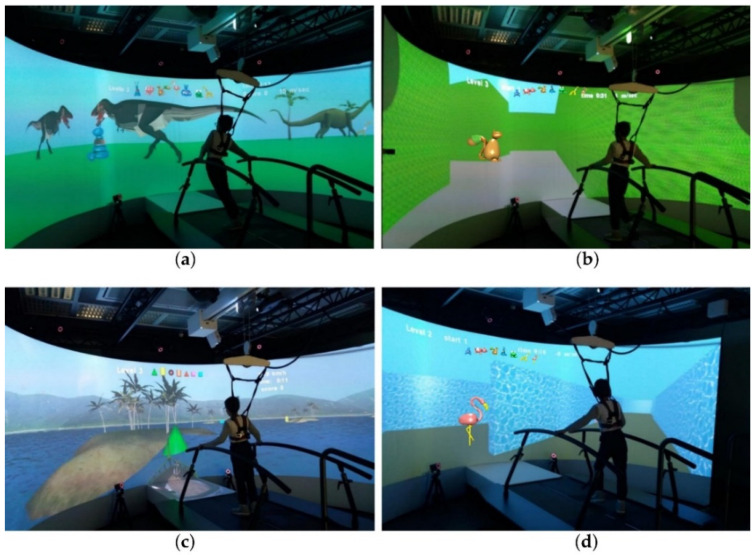
Navigational training applications: (**a**) Dinosaurs, (**b**) Grass maze, (**c**) Boat, and (**d**) Water labyrinth.

**Figure 5 jcm-11-06146-f005:**
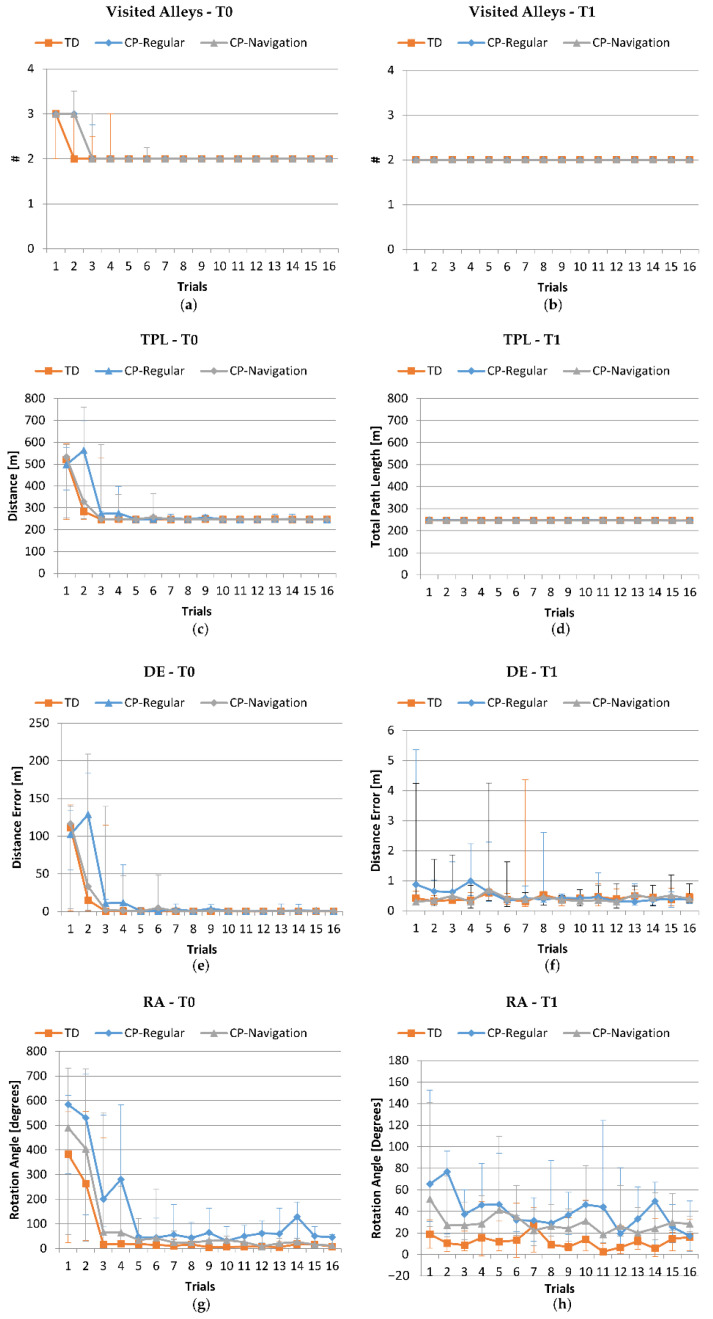
Navigation skills and modifications during the training in the Free Navigational Task evaluated at T0 (left side) and T1 (right side). Learning curves of TD subjects (orange) and participants with CP (CP-Regular: blue, CP-Navigation: grey) were obtained during 16 training sessions. (**a**,**b**) Number of visited alleys (the correct number is 2), at T0 and T1, respectively (#: number of visited alleys); (**c**,**d**) total path length (the minimum is 246 m), at T0 and T1, respectively; (**e**,**f**) distance error (%), at T0 and T1, respectively; (**g**,**h**) rotation angle, at T0 and T1, respectively. Median values and interquartile ranges are reported.

**Figure 6 jcm-11-06146-f006:**
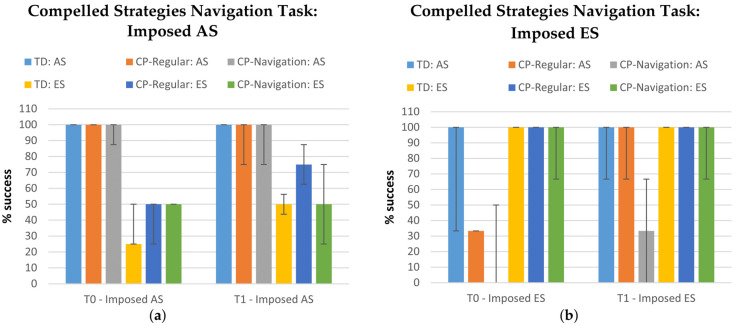
Success (express as percentage) for the trials with imposed allocentric (**a**) and egocentric (**b**) strategies, both at T0 and T1, in TD, CP-Regular, and CP-Navigation participants grouped according to their orienting strategy adopted during the Free Navigation Task (AS and ES subgroups).

**Table 3 jcm-11-06146-t003:** Learning data at the stable trial for TDs (fifth trial at T0, second trial at T1), CPs-Motor (seventh trial at T0, second trial at T1), and CPs-Navigation (seventh trial at T0, second trial at T1), both at T0 and T1.

Learning Parameter		TDs	CPs-Regular	CPs-Navigation	Th. Value	*p* ^&^
Visited Alleys (#)	T0	2 (0)	2 (0)	2 (0)	2	0.506
	T1	2(0)	2(0)	2(0)	2	0.548
	*p* ^$^	0.317	0.655	0.317	-	-
TPL (m)	T0	246.76 (0.71)	252.86 (23.91)	246.67 (2.38)	246	0.677
	T1	246.78 (0.60)	246.85 (1.42)	246.53 (1.39)	246	0.464
	*p* ^$^	0.975	0.286	**0.044**	-	-
DE (m)	T0	0.31 (0.29)	2.79 (9.53)	0.27 (0.97)	0	0.087
	T1	0.32 (0.25)	0.65 (0.70)	0.35 (1.53)	0	0.268
	*p* ^$^	0.875	0.328	0.408	-	-
RA (°)	T0	17.70 (34.53)	56.50 (143.72) *	29.70 (18.04)	0	**0.034**
	T1	10.38 (16.59)	76.79 (77.06) *	27.06 (56.50)	0	**0.019**
	*p* ^$^	0.397	0.248	0.642	-	-

The fourth column (Th. Value) reports the theoretical best value for each parameter. TPL, total path length; DE, distance error; RA, rotation angle. Median (interquartile) values are reported. Significant values of *p* are in bold. ^&^ Value of *p* of the Kruskal–Wallis test. Null hypothesis: the distribution of the learning parameters at T0 (or T1) is the same for CP-Navigation, CP-Regular, and TD groups. * Post hoc analysis shows a statistical different distribution of the RA among CPs-Motor and TDs, both at T0 and T1. Values of *p* in the main text. ^$^ Value of *p* of the Wilcoxon test. Null hypothesis: the distribution of the learning parameters for CP-Navigation, CP-Regular, and TD groups is the same at T0 and T1.

**Table 4 jcm-11-06146-t004:** Distribution of the strategies freely adopted by TD and CP subjects. The table reports the % and the total number of subjects between brackets.

Group		AS	ES	Shifter	No Strategy
CPs-Navigation	T0	50% (7)	36% (5)	14% (2)	0% (0)
	T1	64% (9)	29% (4)	7% (1)	0% (0)
CPs-Regular	T0	18% (2)	46% (5)	27% (3)	9% (1)
	T1	46% (5)	27% (3)	27% (3)	0% (0)
TDs	T0	19% (3)	56% (9)	6% (1)	19% (3)
	T1	56% (9)	31% (5)	13% (2)	0% (0)

AS, allocentric strategy; ES, egocentric strategy.

**Table 5 jcm-11-06146-t005:** Success scores in the trials with imposed allocentric and egocentric strategies, both at T0 and T1, for the whole group of TD, CP-Regular, and CP-Navigation participants.

Test		TDs	CPs-Regular	CPs-Navigation	*p* ^&^
Imposed AS	T0	87.5(43.75)	100(50)	50(25)	0.221
	T1	87.5(43.75)	75(37.5)	75(25)	0.221
	*p* ^$^	0.739	0.157	**0.018**	-
Imposed ES	T0	100(25)	66.7(66.6)	83.33(75)	0.519
	T1	100(33.3)	100(0)	66.7(100)	**0.040**
	*p* ^$^	0.416	0.227	0.310	-

AS, allocentric strategy; ES, egocentric strategy. Median values and interquartile ranges are shown; significant values of *p* are in bold. ^&^ Value of *p* of the Kruskal–Wallis test. Null hypothesis: the distribution of the success scores at T0 (or T1) is the same for CP-Navigation, CP-Regular, and TD groups. ^$^ Value of *p* of the Wilcoxon test. Null hypothesis: the success score distribution for CP-Navigation, CP-Regular, and TD groups is the same at T0 and T1.

## Data Availability

The data presented in this study are openly available in Zenodo at https://doi.org/10.5281/zenodo.6536242 (accessed on 10 May 2022). The “Star-Maze” application presented in this study is openly available in Zenodo at 10.5281/zenodo.5457559 (link: https://zenodo.org/record/5457559#.YVXgXbUzZPY, accessed on 5 September 2021).

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
