# Peer review of "Could an Immersive Virtual Reality Training Improve Navigation Skills in Children with Cerebral Palsy? A Pilot Controlled Study"

_jcm, 2022, doi:10.3390/jcm11206146_

Round 1

Reviewer 1 Report

Manuscript ID#: JCM-1916362

Title:
Could an Immersive Virtual Reality training improve navigation skills in children with Cerebral Palsy? A pilot controlled study

Authors: R. Nossa, et al.

This manuscript introduces using a virtual reality (VR) training to help improving the navigation skills in children with Cerebral Palsy (CP). The experiment results based on 35 patients and 14 typically developing controls showed that VR training appeared to be effective for this purpose. The work is interesting in providing new insight into the patients with CP. Some concerns are as follows:

1.       The VR system used the GRAIL system with an 180-degree display surrounding a treadmill for gait analysis in a room setting. In fact, this is a half-immersive VR setting. A completely immersive VR usually requires a headset / goggle entirely blocking views of all other irrelevant objects, in which the immersive experience is very different from such a half-open VR setting. Therefore, it may not be appropriate to name this GRAIL-based VR setting in this manuscript as immersive VR (IVR).

2.        A number of key elements are not described clearly. Although may have been introduced or described elsewhere in the cited papers, these elements are expected to be briefly outlined in the current manuscript. For example, in the navigation tasks, why starting from alley 4 and finding the treasure in alley 3 is using AS, and finding in alley 1 is using ES is not clear.

3.       P4, section 2: why two tasks, i.e., one regular and one navigation, were needed is not exactly clear. It appears later in the result section (P.11) that these 2 were assigned to CP patients with different levels of GMFCS (I vs. II & III) functions. If this was true, the information should be mentioned earlier in the Material and Method section, probably in Section 2.1.

4.       Figure 1 is nice in providing an overview of the study design. Again, the subgrouping strategy of CP-regular and CP-navigation is not mentioned, and presenting suddenly 11 and 16 participants in each subgroup makes the manuscript odd to follow.

5.       Page 10, the strategy adopted by the participants included allocentric, egocentric, no-strategy. Although a reference 12 is cited here, it is preferred that brief explanations are also presented concerning the 3 important strategies.

6.       P6, it seems that the navigation task involved two types of navigation sub-tasks  -- the free and compelled strategies. In addition, 3 strategies were included in the latter strategy. The text is not exactly clear on these points, and more details concerning the corresponding theory, concepts/idea and the corresponding design in support of such a set-up are expected here. Please elaborate.

7.       Page 6, last paragraph: the description is confusing. In the 21 times of search, 16 attempts were meant to let the participant to learn to get the treasure; and 5 interposed trials were to assess their AS/ES strategy in which the participant was also to find the treasure. Does this mean that all the 21 searches would end up with getting a treasure in each trial? Section 2.5.2 mentions “the success or no success in finding the reward” infers that not every trail may get a reward. Please clarify.

8.       Following pint #7, was a success or a failure simply judged based on the procedure how the participant got the treasure, but not the result whether or not the participant got the treasure? The text also mentions later that “… successful trails were those with a direct route from the starting point to the alley with the reward”. The definition of “success” or “successful” is therefore confusing. Please clarify.

9.       P7, 2nd paragraph, “during the four compelled AS trials…..while in the three compelled ES trials”: why there were 4 and 3 trials here is not clear. The information appears abrupt in the context.

10.   P7, Section 2.4.1 the regular IVR training program:  technical details about the VR system are not clear. The text mentioned “improve the balance”, but how? Does it mean that displaying irregular ground and letting the participant completing the task would certainly improve the balance? Something is missing here. Please rewrite.

11.   Same place, “…while catching objects with their hands…”: no information in the text has previously described the related VR setting. Does the VR system contain a digital glove or a handheld device to allow the participant “catch objects”, or in fact it actually simply means the participant was just holding something irrelevant to the experiment during the task? More technical details are needed here for readers to understand the experiments.

12.   P8, the third paragraph, “…rescue the animals…. lead them to a water fall….”In the VR setting, it is not clear how the participant acted to “rescue” and “lead”. Technical details concerning user-computer interaction are needed.

13.   Section 2.5.1., “Z-scores were computed….” Not clear whether the authors actually wanted to say that the z-scores were computed based on comparisons between the groups? Please revise.

14.   P10, “… stabilization trials ….as the 5th and 7th ones”, then how the data before the 5th or 7th trials were treated? Were the data abandoned or anyway included in the analyses without discriminations?

15.    Finally, the half-open environment and the displayed scenes in the VR setting were quite primitive, which may not provide to the participants the feeling of “real”, and may thus introduce extra disturbance. New systems such as the Google Oculus provide much more realistic and a totally immersive environment. While the work is unable to be moved to a new technology platform right away, it certainly should be mentioned as a limitation of the current work.

Minor points:

16.   P1, Abstract: the acronym CP used before they were defined

17.   P16, Figure 5 seems missing a part in the top.

Reviewer 2 Report

The purpose of this manuscript was to study navigation and spatial skills pre and post  intervention in children with CP who performed an IVR training program and to compare the skills of the group that underwent a landmark-based training for half of training time with respect to children with CP who did not. This manuscript reflected work that has been completed. However, few points still need to be clarified

Page 2 of 22, Line 82-83.  The reference of work of Biffi and collaborators should be provided.

Page 2 of 22, Line 77-82.  Authors mentioned about non-immersive VR system and immersive VR(IVR).  However, more details of IVR should be added in this paragraph.  IVR has two major forms of delivery. The first is a headmounted
display (HMD) that offers wide field-of-view stereoscopic vision. The second is a projection-based system—a computer-assisted virtual environment (Cave) whose walls are back-projection stereo display screens.  In this study, authors used the second forms.  

Page 10 of 22, Line 372-373  “…subjects were classified in ….. efficient strategy (no-strategy hereafter)”.  Who will do these classifications, and in which manner?  By onsite observation or review it by videotape?

Page 11 of 22, Table 1.  In the first column, RAVEN Z-scores should be replaced by “Raven’s Colored Progressive Matrices” since these data indicated the results of this test.

Page 12 of 22, Table 2.  The p-value in the Corsi block test Z-score at T0 should be in bold since this value is 0.032<0.05

Page 17 of 22, Line 626.  “…CP Navigation participants could rely on the e flexible representations”.  There was a typographical error in this line.

Page 17 of 22, Line 629~640.  Could authors provide explanations to illustrate why use of allocentric strategy improve navigation skills in children with Cerebral Palsy

In virtual reality (VR), especial in IVR system, users often experience symptoms of motion sickness, which is referred to as VR sickness or cybersickness. Did any participants in this study report such side effects?

Round 2

Reviewer 1 Report

Manuscript ID#: JCM-1916362.R1.

Title:
Could an Immersive Virtual Reality training improve navigation skills in children with Cerebral Palsy? A pilot controlled study

Authors: R. Nossa, et al.

The revision has satisfactorily addressed most of the concerns and now it is much clearer in presenting the work, although there is a few location indications of the revised text in the  revision was not correct (e.g. answers to Q3, Q11, Q 12). Two minor points left:

 1. Q13, and in the text L402-405 (Section 2.5.1): the revised text does not read correct. Please double check and update.

  2. The answers to the questions have well addressed the concerns. Please be sure the answers are also included in the text, as I believe that these will also be potential readers’ concerns.
